# The Response of Antioxidant Enzymes and Antiapoptotic Markers to an Oral Glucose Tolerance Test (OGTT) in Children and Adolescents with Excess Body Weight

**DOI:** 10.3390/ijms242216517

**Published:** 2023-11-20

**Authors:** Maria Efthymia Katsa, Eirini Kostopoulou, Tzortzis Nomikos, Anastasios Ioannidis, Vasileios Sarris, Spyridon Papadogiannis, Bessie E. Spiliotis, Andrea Paola Rojas Gil

**Affiliations:** 1Laboratory of Basic Health Sciences, Department of Nursing, Faculty of Health Sciences, University of Peloponnese, 22100 Tripoli, Greece; merfykat@hua.gr (M.E.K.); tasobi@uop.gr (A.I.); vsarris@di.uoa.gr (V.S.); spirospapadogiannis@gmail.com (S.P.); 2Department of Nutrition and Dietetics, School of Health Sciences and Education, Harokopio University, 17676 Athens, Greece; tnomikos@hua.gr; 3Division of Pediatric Endocrinology, Department of Pediatrics, School of Medicine, University of Patras, 26504 Patras, Greece; eirini.kost@gmail.com (E.K.); besspil@gmail.com (B.E.S.)

**Keywords:** impaired glucose tolerance, children, overweight, oxidative stress, apoptosis

## Abstract

Oxidative stress and apoptosis are involved in the pathogenesis of obesity-related diseases. This observational study investigates the antioxidant and apoptotic markers response to an oral glucose tolerance test (OGTT) in a population of overweight children and adolescents, with normal (NGT) or impaired glucose tolerance (IGT). Glucose, insulin, and C-peptide concentrations, as well as oxidative stress (SOD, GPx3) and apoptotic markers (Apo1fas, cck18), were determined at T = 0, 30, 60, 90, 120, and 180 min after glucose intake during OGTT. The lipid profile, thyroid function, insulin-like growth factor1, leptin, ghrelin, and adiponectin were also measured at baseline. The 45 participants, with a mean age of 12.15 (±2.3) years old, were divided into two subcategories: those with NGΤ (n = 31) and those with IGT (n = 14). The area under the curve (AUC) of glucose, insulin, and C-peptide was greater in children with IGT; however, only glucose differences were statistically significant. SOD and GPx3 levels were higher at all time points in the IGT children. Apo1fas and cck18 levels were higher in the NGT children at most time points, whereas Adiponectin was lower in the IGT group. Glucose increased during an OGTT accompanied by a simultaneous increase in antioxidant factors, which may reflect a compensatory mechanism against the impending increase in oxidative stress in children with IGT.

## 1. Introduction

Over the last few decades, the prevalence of childhood obesity and its consequences have significantly increased globally [1]. Obesity is associated with adverse health issues leading to metabolic syndrome [2]. Amongst them, insulin resistance, expressed as disorders of insulin metabolism, including impaired fasting glucose (IFG), impaired glucose tolerance (IGT), and type 2 diabetes mellitus (DM2), is implicated in microvascular and macrovascular complications occurring from early childhood [3]. Τhe identification of the presence of glucose intolerance in overweight or obese children is of major importance. For this purpose, the performance of an oral glucose tolerance test (OGTT) has been suggested in children with obesity [4,5]. Our team has recently reported that in children with obesity who continue to gain weight, a second OGTT may also be necessary in order to diagnose IGT [6].

Growing evidence suggests that oxidative stress plays a major role in the pathogenesis of obesity-related diseases. Children and adolescents with obesity and metabolic syndrome exhibit disruption in the critical oxidant–antioxidant equilibrium [7]. The imbalance involves increased production of reactive oxygen (ROS) and nitrogen (RNS) species and inefficiency of enzymatic (catalase, glutathione peroxidase (GPx), and superoxide dismutase (SOD)) and non-enzymatic (glutathione and uric acid) antioxidative systems [8,9]. The increased levels of oxidative stress markers, such as homocysteine and malondialdehyde, have been reported in children with obesity, as opposed to reduced levels of antioxidants, i.e., SOD and glutathione peroxidase activities [10,11]. Similarly, increased levels of nitrite/nitrate and decreased levels of glutathione peroxidase have been observed in children with obesity and insulin resistance, including a positive association between the nitrite/nitrate levels and Homeostasis Model Assessment—Insulin Resistance (HOMA-IR), which is a surrogate marker of insulin resistance, and a negative association between glutathione peroxidase and HOMA-IR [12]. Homeostasis Model Assessment—Insulin Resistance has also been found to be negatively associated with lipid-corrected beta-carotene, serum alpha-tocopherol levels, and SOD and catalase activity in the erythrocytes of non-obese children [13]. Oxidative stress is considered a key factor in the pathophysiology of insulin resistance due to its effect on insulin receptor signal transduction [14]. It has also been associated with hypertension in a multi-ethnic population of children and adolescents with obesity [15], as well as with non-alcoholic fatty liver disease [16].

Furthermore, poor glycaemic control has been linked with reduced antioxidant activity in the paediatric population [17]. Evidence also suggests that nitric oxide (NO) release and bioavailability are reduced in the insulin-resistant state, leading to endothelial dysfunction [18]. Oxidized Low-Density Lipoprotein (LDL) is associated with insulin resistance, independent of the presence of obesity, and it has been hypothesized that oxidative stress may be independently related to the development of insulin resistance from an early age [19,20].

Superoxide dismutase and GPx constitute important anti-oxidant factors against oxidative stress [21]. Superoxide dismutase catalyzes the dismutation of superoxide anion free radicals (O2^•−^) into molecular oxygen and hydrogen peroxide (H_2_O_2_), and by eliminating O2^•−^, which restricts cell damage [22]. Glutathione peroxidases catalyze the reduction in hydrogen peroxide or organic hydroxyperoxides using glutathione. Glutathione peroxidase 3 is among the most well-studied GPxs. It is highly expressed in the adipose tissue and it has been demonstrated that its expression is reduced in the serum and adipose tissue of subjects with obesity. It has been proposed that increased oxidative stress in obesity is associated with diminished adipose GPx3 expression and reduced circulating GPx3 [23].

In addition to oxidative stress, Apo1fas, an important death receptor that belongs to the TNF/receptor family, is increased in adults with obesity, DM2, hypertension, and cardiovascular disease, and in the liver of patients with non-alcoholic fatty liver disease [24,25]. It is believed that Apo1fas contributes to the attraction of immune cells to adipose tissue through elevated Fas-induced chemokine production. In hypertrophic adipocytes, it is expressed at higher levels compared to smaller adipocytes [26,27]. Apo1fas also appears to be associated with the lipid profile. A study conducted in a healthy adult population showed that Apo1fas was positively correlated with triglycerides and negatively correlated with High-Density Lipoprotein (HDL). Hence, it is believed that adiposity is related to apoptosis or tissue damage; however, the exact mechanism remains unknown [28]. In children, the role of apoptosis has not yet been clarified. A recent study indicated that Apo1fas and apoptosis in general may have a protective role in children’s predisposition to metabolic syndrome through the down-regulation of inflammation and atherosclerosis; however, in adults, it has been shown to contribute to atherosclerosis [29].

Cytokeratin 18 (cck18) is a protein that belongs to the intermediate filament group and is present in most epithelial and parenchymal cells. During apoptosis, it is cleaved by the action of caspases and released into the blood circulation as caspase-cleaved CK-18 (CCCK18 or cck18) [30]. Several studies have demonstrated that cck18 plasma concentrations are increased in patients with different forms of carcinomas, including breast, prostate, colon, lung, head and neck, testicular, pancreatic, and gastrointestinal adenocarcinoma [31,32,33,34,35,36]. Cck18 has also been characterized as a possible serological marker of epithelial apoptosis in adult patients with obesity or type 2 diabetes mellitus. Moreover, it is considered a marker of hepatic function, which is increased in liver steatosis in type 2 diabetes mellitus [37].

The purpose of the present study was to investigate the response of antioxidant markers, such as SOD and GPx3, and of apoptotic markers, such as Apo1fas and cck18, during an OGTT in a paediatric population with overweight or obesity, taking into consideration disorders of glucose metabolism and biochemical parameters.

## 2. Results

### 2.1. Descriptive Characteristics

The majority of the participants, 60%, were male. The mean age of the participants was 12.15 ± 2.3 years (12.79 ± 1.88 years old for the IGT group and 11.87 ± 2.5 years old for the NGT group) (Table 1).

Children and adolescents with abnormal glucose metabolism had significantly higher glucose values (*p* < 0.05), insulin values (*p* < 0.05), and HOMA-IR (*p* < 0.05), and significantly lower adiponectin values (*p* < 0.05) compared to those with normal glucose metabolism. There were no significant differences in the other studied marker, between the NGT and IGT children (Table 1).

### 2.2. Glucose, Insulin, and C-Peptide Response during the OGTT

Figure 1, Figure 2 and Figure 3 depict glucose, insulin, and C-peptide response curves during the OGTT, and the corresponding AUCs in the participants with NGT and those with IGT. The changes in the glucose levels during the OGTT exhibit different patterns in the two studied categories. Children and adolescents with IGT exhibited higher glucose levels at all time points. The highest glucose levels were observed at T = 60 min, whereas in children with NGT, at T = 30 min. The glucose levels at T = 180 min were higher compared to those at the baseline, particularly in the IGT group.

With regard to the insulin levels, the pattern of change was similar between children with NGT and children with IGT; however, the levels were higher in children with ΙGΤ at all time points. Insulin levels peaked at T = 30 min and T = 120 min in both groups. In both studied categories, the insulin levels at T = 180 min were higher than those at T = 0.

Regarding the C-peptide levels, a peak was noted at T = 60 min and T = 120 min in both categories. At T = 180 min, C-peptide was higher than the baseline (T = 0 min) in both groups. Both insulin and C-peptide curves were biphasic, as opposed to the monophasic curve of glucose.

The AUCs of glucose, insulin, and C-peptide were greater in children with IGT; however, only the glucose differences were statistically significant.

### 2.3. Antioxidant Markers (SOD, GPx3) during the OGTT

Figure 4 and Figure 5 depict the SOD and GPx3 responses during the OGTT, as well as the corresponding area under the curve (AUC), in the two studied populations. SOD and GPx3 levels were significantly higher at all time points in the IGT children compared to the NGT children. During the OGTT, the pattern of change in the antioxidant enzymes differed between the IGT and the NGT children. In the IGT children, the SOD levels gradually increased from the baseline to T = 90 min when they reached their maximum levels and subsequently decreased. At T = 180 min, the SOD levels were slightly higher than those at T = 0 min. In the NGT children, SOD levels decreased from T = 0 min to T = 30 min and then increased till T = 90 min when they reached their highest levels. The variability of the SOD levels was greater in the IGT children compared to the NGT children.

The GPx3 levels increased in the IGT children from T = 0 min to T = 60 min, and then remained stable till T = 90 min and decreased from T = 90 min till the end of the test. In the NGT children, GPx3 levels increased slightly from T = 0 min to T = 60 min and then gradually decreased, exhibiting very small variability.

The AUCs of SOD and GPx3 were larger in the IGT group, but the difference was not statistically significant.

### 2.4. Apoptotic Markers (Apo1fas, cck18) during the OGTT

Figure 6 and Figure 7 depict Apo1fas and cck18 responses during the OGTT, as well as the corresponding AUCs, in children of the two studied categories. Apo1fas levels were lower in the IGT children at most time points, with the exception of T = 90 min. In the IGT children, Apo1fas exhibited two low peaks (at T = 30 min and at T = 120 min) and one high peak (at T = 90 min), and finally reached similar levels at T = 180 min as at T = 0 min. In the NGT children, Apo1fas levels increased from T = 0 min to T = 60 min, decreased again to reach the baseline levels at T = 90 min, and remained stable thereafter. The AUC of Apo1fas was larger in the participants with NGT compared to the participants with IGT, but the difference was not statistically significant. 

In the IGT children, cck18 varied significantly during the OGTT. Two high peaks were observed at T = 60 min and T = 120 min, and three low peaks were observed compared to the baseline at T = 30 min, T = 90 min, and T = 180 min. In contrast, cck18 levels showed low variability among the different time points in the NGT children. A slight decrease was observed from T = 0 min (highest levels) to T = 180 min (lowest levels). The AUC of cck18 was lower in the participants with NGT compared to the participants with IGT, but the difference was not statistically significant.

### 2.5. Baseline Correlation Analysis

#### 2.5.1. Leptin, Ghrelin, and Adiponectin Levels

No statistically significant differences were found in the leptin and ghrelin levels between children and adolescents with IGT and those without. Adiponectin levels were significantly lower in children and adolescents with IGT compared to the NGT children and adolescents (Table 1).

#### 2.5.2. Linear Regression Analysis in Children with Normal OGTT

Linear regression analysis showed statistically significant correlations, as shown in Table 2.

#### 2.5.3. Linear Regression Analysis in Children with Abnormal OGTT

Linear regression analysis showed statistically significant correlations, as shown in Table 3.

## 3. Discussion

Paediatric obesity has been shown to follow a worrisome rise worldwide, particularly in Greece. According to one of the largest nationwide studies conducted on over 330,000 children and adolescents in Greece, obesity was identified in 9% of boys and 7.5% of girls [38]. Another study from Peloponnese in Southern Greece showed that 16.6% of the boys and 10.5% of the girls were obese, and 13.5% of the boys and 16.7% of the girls were overweight [3]. The most recently published studies reporting the prevalence of paediatric overweight and obesity in Greece demonstrated higher prevalence rates compared to previous national estimates. Specifically, among 10- to 12-year-old children and adolescents, the prevalence of overweight and obesity was 25.9% and 12.6%, respectively, whereas among 10- to 16-year-old children and adolescents, the corresponding rates were 19.2% and 15.6%, respectively [39,40].

Obesity, disturbed glucose metabolism, oxidative stress, and apoptosis, closely co-exist. The association between excess body weight or abdominal obesity and insulin resistance in childhood and adolescence is well established, and possible mechanisms involve macrophage infiltration of the adipose tissue and increased production of pro-inflammatory cytokines, adipokines, and fatty acids in the blood secondary to chronic low-grade inflammation [41,42].

Adipocyte apoptosis is prominent in adults with obesity, and a positive relationship has been found between adipocyte apoptosis and markers of insulin resistance. Apoptosis, also referred to as programmed cell death, is essential for normal development and homeostatic death in all multicellular organisms [43]. Reactive oxygen species (ROS) and the resulting oxidative stress seen in subjects with obesity play a pivotal role in apoptosis. Although ROS have beneficial roles at low concentrations by acting as a second messenger in several signal transduction pathways, including those of growth factors and cytokines [44], when they are produced in excess, their production overwhelms the biochemical defenses of the cell, resulting in severe cell injury [45]. Furthermore, oxidative stress impairs mitochondrial function, leading to a significant decrease in the cellular energy supply, thus contributing to apoptosis. In contrast, Bcl-2, an important oxidative stress marker, is known to have an anti-apoptotic action in multiple cell tissues [46].

In our present study, it is of particular interest that SOD and GPx3 levels were significantly higher during the OGTT in children and adolescents with impaired glucose tolerance compared to children with normal glucose tolerance at all time points. Also, contrary to the findings of previous studies in adults [47], in children with IGT, SOD and GPx3 levels increased in parallel to the glucose excursion (from T = 0 min to T = 30 min), but remained increased for a longer time (till T = 90 min) compared to the glucose concentrations (T = 60 min). In contrast, in children with normal glucose tolerance (NGT), the observed dynamics of SOD differed from the IGT group. In the NGT children, GPx3 followed the same pattern of change as in the IGT children; however, glucose concentrations showed a smaller variability. It is possible that the above findings may suggest a protective role of SOD and GPx3 in children with impaired glucose metabolism, who perhaps recruit antioxidant defensive mechanisms against the rise in glucose concentrations and the oxidative stress-mediated consequences.

The relationship between the antioxidant enzymes, SOD and glutathione peroxidase, and glucose excursions during an OGTT has previously been investigated. In a study conducted in 36 healthy adults and 36 adults with metabolic syndrome, it was found that both SOD and glutathione peroxidase activity were decreased at T = 120 min of an OGTT compared to T = 60 min in both groups [48]. In contrast, other studies have also shown that glucose excursion during an OGTT was correlated with an increase in oxidative stress markers and a decrease in antioxidants in adults with normal glucose tolerance, impaired glucose tolerance, or type 2 diabetes mellitus [49].

Our results also showed a negative correlation between Apo1fas and BMI% and a positive correlation between Apo1fas and apolipoprotein-alpha in children with normal OGTT. Apo1fas AUC was also positively correlated with apolipoprotein-alpha in children with impaired glucose tolerance. Therefore, there is a clear need to further investigate the role of apoptosis, especially of Apo1fas, in the metabolism of children with normal and impaired OGTT. It is possible that the homeostatic mechanisms present in children act differently in these two categories. Interestingly, a previous study carried out in healthy children demonstrated the induction of apoptosis of cells involved in atherosclerosis as well as the promotion of non-inflammatory clearance of the apoptotic cells, particularly in younger children, suggesting a potential protective role of apoptosis [29].

In addition, in a large cross-sectional Chinese survey, low apolipoprotein-alpha was associated with an increased risk of pre-diabetes, insulin resistance, and hyperinsulinaemia [50]. Concerning the relationship between apolipoprotein-alpha and apoptosis, it has been shown that its expression causes a decline in survivin protein expression, an inhibitor of apoptosis, and a cell cycle promoter [51]. Apolipoprotein α1 has been found to improve pancreatic β-cell function. The mechanism linking low apolipoprotein α1 to insulin resistance has not been fully elucidated. However, the hypothesis that apolipoprotein α1 has a beneficial effect on the vascular endothelium has been documented. In particular, it prevents endothelial apoptosis caused by oxidized LDL and TNFα. Its anti-inflammatory effect has also been proven, due to which insulin resistance may be reduced [52].

With regard to the apoptotic Apo1fas and cck18 markers, their levels exhibited a negative correlation with glucose concentrations during the first 30 min of the OGTT in children with IGT. Also, after the glucose peak at T = 30 min, both Apo1fas and cck18 started increasing, reaching a peak at T = 90 min and at T = 60 min, respectively. Subsequently, the apoptotic markers decreased again, following the peak of the antioxidants, SOD and GPx3. These findings may suggest a counter-regulatory action of the antioxidant markers in an attempt to protect against the glucose-mediated oxidative stress during the OGTT in obese children with IGT. This has been described in young patients with type 1 diabetes mellitus (DM1) [53]. Furthermore, a negative correlation between insulin and Apo1fas response curves was noted in children with IGT, which is in agreement with the finding that the Apo1fas AUC was negatively correlated with insulin in these children. In the same children (IGT), cck18 showed a similar pattern during the OGTT, with the same peaks (at T = 60 min and T = 120 min) as C-peptide. Also, cck18 AUC was negatively correlated with glucose levels in children with IGT. Similarly, a previous survey also showed a negative correlation between Apo1Fas and fasting glucose in children, even in the presence of obesity, hypothetically reflecting a protective mechanism against predisposing factors of atherosclerosis and metabolic dysfunction through reduced apoptosis [29].

Our results also show that in the NGT group, Apo1fas increased parallel to glucose levels, exhibiting a more gradual and longer duration (till T = 60 min) increase compared to glucose levels. Cck18 showed a gradual decrease throughout the OGTT, as well as low variability. Additionally, in children with normal OGTT, the AUC of Apo1fas was positively correlated with the AUC of GPx3. Apo1fas was also negatively correlated with HbA1c in children with normal OGTT, which possibly suggests that Apo1fas distribution after glucose loading may have a protective role in glycemic control. In contrast, the cck18 AUC was positively correlated with insulin, HOMA-IR, and HbA1c levels in the NGT group. Caspase-cleaved CK-18 is implicated in liver impairment in children and adolescents with DM1 and in adult patients with DM1 and DM1 with insulitis. It appears that cck18 could be correlated with a predisposition to metabolic disorders and DM2 in children with obesity and normal glucose tolerance test results [37,54]. The present study is the first to investigate the potential diagnostic and prognostic significance of cck18 in children with impaired glucose metabolism. Further research in this field, however, is necessary.

Another interesting finding of our study is that leptin and ghrelin levels did not differ significantly between the children and adolescents with IGT and those without, as opposed to adiponectin levels that were lower in the IGT group. Leptin regulates fat and glucose metabolism [55]. Among children with obesity, increased leptin levels, attributed to leptin resistance, are well documented. Despite high levels, its action may be impeded by leptin resistance [56]. Leptin has also been associated with insulin resistance and it may be involved in its pathogenesis, [57,58]; however, this association was not supported by our findings. In addition, ghrelin, an orexigenic hormone, which increases before meals and decreases after meals, has also been reported to decrease after glucose is administered during an OGTT. Additionally, it has been found to follow an inverse pattern compared to insulin during a 24 h observation in normal subjects [59,60]. In children with obesity, an inverse relationship between fasting ghrelin, insulin levels, and insulin resistance has also been reported [61]. However, this was not confirmed by our results. Adiponectin levels have been reported to be significantly lower in children with obesity compared to non-obese children. An inverse correlation between adiponectin and obesity and insulin resistance has also been documented in both sexes during puberty [62]. This finding is consistent with our findings that children with excess weight and IGT have lower fasting adiponectin levels than those with excess weight and NGT.

Our study has some limitations, including the small sample size and the fact that the response curves of the antioxidant and apoptotic markers during the OGTT were not correlated with the degree of excess weight (overweight versus obesity), in order to avoid further reducing the comparative sample size. Furthermore, all patients with insulin resistance had impaired glucose tolerance, and one also had impaired fasting glucose; hence, no comparisons were made between children and adolescents with IGT, IFG, and DM2. Finally, considering the ethical issue that performing an OGTT in normal-weight children would be inappropriate, no control group was included in the study.

## 4. Material and Methods

### 4.1. Subjects and Ethics

Forty-five (45) children and adolescents (27 boys and 18 girls), aged 7 to 16 years, were recruited for this observational study from the Outpatient Clinic of the Department of Paediatric Endocrinology of the University General Hospital of Patras in Greece. The selection of the participants was random and unbiased. Informed consent was obtained from the parents or guardians of the participants after they were informed about the purpose and the procedure of the study. The participation was voluntary and the anonymity of the participants and their parents/guardians was ensured. The study was approved by the Research Ethics Committee of the University General Hospital of Patras (25,860), Medical School, University of Patras, and was conducted in accordance with the Helsinki Declaration of 1975, as revised in 1983. All participants were overweight or obese and did not take any medication. None of the subjects had any significant medical illnesses at the time of the study. Any other health issue was excluded.

### 4.2. Anthropometric Measurements

A full physical examination of all the children was performed. Anthropometric measurements were obtained, including children’s height and weight. Height was measured using a wall-mounted stadiometer (Seca 216, Hamburg, Germany). Body weight was measured between 9 a.m. and 10 a.m. using a standard beam balance scale (Seca Supra 719, Hamburg, Germany) while the subjects wore no shoes and light clothing. Body mass index (BMI) was calculated by dividing the weight in kilograms by the square of the height in meters. The BMI percentile for each subject was estimated using the actual BMI for age and sex based on data from the 2000 Greek growth charts. The cut-off point for obesity proposed by the IOTF corresponded to BMI values at the 90th percentile for boys and the 95th percentile for girls [63,64].

### 4.3. OGTT

A 3 h OGTT was performed on all 45 children after a 12 h fasting period. The dose of the administered glucose was 1.75 g/kg of body weight, with a maximum dose of 75 g. Blood samples were obtained before glucose was administered (T = 0 min), and 30 min (T = 30 min), 60 min (T = 60 min), 90 min (T = 90 min), 120 min (T = 120 min), and 180 min (T=180 min) after glucose was given. Glucose, insulin, and C-peptide concentrations, as well as oxidative stress and apoptotic markers, were determined from the blood samples. A 3 h OGTT was performed because, in our experience, it can not only identify children who have an abnormal glucose response at the 2 h point but also those who are able to normalize their glucose levels at the 3 h time point due to a delayed insulin response.

In order to better understand the role of disturbed glucose metabolism in the processes of apoptosis and oxidative stress, the studied population was divided into two categories: participants with normal glucose tolerance during the OGTT (NGΤ, n = 31) and participants with impaired glucose tolerance during the OGTT (IGT, n = 14). IGT was defined according to the American Diabetes Association guidelines as a two-hour plasma glucose level of 140 to 200 mg/dL [65]. Among the participants with IGT, one also had Impaired Fasting Glucose (IFG), which was defined as a fasting plasma glucose level of above 100 mg/dL, but less than 126 mg/dL. None of the participants met the criteria for DM2 (fasting plasma glucose level of above 126 mg/dL or two-hour plasma glucose levels of above 200 mg/dL). The BMI of the two studied categories did not differ significantly.

### 4.4. Laboratory Tests, Blood Samples, and Serum Assays

Before the initiation of the test (T = 0 min), peripheral blood samples were obtained for biochemical investigations. Glucose (Olympus Diagnostica, Hamburg, Germany), insulin (Sorin Biomedica, Sallugia, Italy), C-peptide (by radioimmunoassay Merck KGaA, Darmstadt, Germany), HbA1c (Zivak Hemoglobin A1c HPLC Analysis Kit), lipid profile (total cholesterol, HDL, LDL, triglycerides) (Olympus Diagnostica, Hamburg, Germany), markers of thyroid function (T3, T4, TSH, FT4) (DiaSorin, Sallugia, Italy) and Insulin-like growth factor 1 (IGF-1) (Immunotech, Czech Republic) were measured. Leptin, ghrelin, and adiponectin were also measured using Elisa (Phoenix pharmaceuticals, Belmont, CA, USA). All serum analyses were conducted at the same lab following the same procedure.

### 4.5. Studied Indexes

The marker HOMA-IR was calculated using the following formula: HOMA-IR = Fasting Insulin (μUI/mL) × Fasting Glucose mmol/L/22.5 (23). The insulinogenic index, an indicator of β-cell function, was calculated as follows: (Insulin 30′ − Fasting insulin 0′)/(Glucose 30′ − Fasting Glucose 0′) [66]. The area under the curve (AUC) for glucose, insulin, and C-peptide was calculated using the trapezium rule [67].

### 4.6. Antioxidant Markers

#### 4.6.1. Superoxide Dismutase 3 (SOD3)

The determination of the activity of SOD3 in the serum was based on the classical protocol described by McCord and Fridovich. The activity of SOD3 was assessed by the inhibition of superoxide anion-induced ferricytochrome c oxidation. Superoxide anion is produced endogenously by the reaction of xanthine with oxygen, which is catalyzed by xanthine oxidase. Briefly, an assay cocktail containing 50 mM PBS (pH 7.8)/0.1 mM EDTA, 0.5 mM xanthine, and 0.01 mM cytochrome C (ferricytochrome c) (Sigma Aldrich, St. Louis, MO, USA) was placed in a cuvette. A solution of xanthine oxidase (~3 mIU) (Sigma Aldrich, St. Louis, MO, USA) was then added and the absorbance at 550 nm was recorded every 30 s for 3 min using a Spectrophotometer Novaspec II (Pharmacia Biotech, Apeldoorn, The Netherlands). The added amount of xanthine oxidase should increase the rate of absorbance between 0.015 and 0.025. The reaction was repeated with the addition of an aliquot of serum. The amount of SOD3 in the serum should result in a 40–60% inhibition of the rate of increase in absorbance. The number of units of SOD in the sample assay was calculated using the following formula: units = % inhibition/(100 − % inhibition) [68].

#### 4.6.2. Glutathione Peroxidase 3 (GPx3)

The determination of GPx3 was based on the method of Wendel (1980) adapted for microwell plates [69]. According to this method, GPx3 catalyzes the reduction of H_2_O_2_^−^ to water in the presence of reduced glutathione (GSH), which is converted to its oxidized form (GSSG). Oxidized glutathione is then recycled to GSH in the presence of NADPH in a reaction catalyzed by glutathione reductase. The rate of NADPH reduction in the reaction mixture was proportional to the GPx3 activity. Briefly, serum samples (10–30 μL) were added to a reaction mixture containing phosphate-buffered saline (PBS) 50 mM/EDTA 0.4 mM/pH 7.0/sodium azide 1 mM, (PBS/Sodium azide), glutathione reductase (1 U/mL), GSH (1 mM), and NADPH (0.12 mM) (Sigma Aldrich, St. Louis, MO, USA). The reaction was initiated by the addition of H_2_O_2_ (0.00084%) to the microwell plates. The absorbance of NADPH at 340 nm was recorded every min for a total of 7 min using a microplate reader (PowerWave XS, BioTek, Winooski, VT, USA). The activity of GPx3 in U/mL was calculated using the following formula: [(ΔA/min,sample − ΔA/min,blank) × 0.5]/[4.6 × Vsample] [70].

### 4.7. Apoptotic Markers

Two apoptotic markers, APO1/Fas (Human Fas/TNFRSF6/CD9 5 ELISA Kit NBP1-91190, Novus Biologicals, Centennial, CO, USA) and cck18 (caspase-cleaved Keratin 18) (ELISA kit ALX-850-270-KI01, Enzo Life Sciences, Peviva, NY, USA), were measured using ELISA.

### 4.8. Statistical Analysis

Normally distributed continuous variables are presented as mean ± standard deviation; normality was evaluated using the Kolmogorov–Smirnov criterion. Categorical variables were presented as frequencies. Differences in the mean values of the subjects’ baseline characteristics between the different intervention groups were assessed by analysis of variance (ANOVA) after checking for homoscedasticity using the Levene test. Repeated measures analysis of variance (RM-ANOVA) was used to compare the postprandial curves of the biochemical parameters between the group of children with NGT and the group of children with IGT. Differences between groups in summary measures in the postprandial state [areas under the concentration-versus-time curves (AUC), calculated by using the trapezoidal rule] were evaluated by using the Student’s t test. Baseline correlation analysis was performed between the studied markers and other possible confounding parameters, specifically gender and age. Correlation analysis was also performed between the AUC of the studied markers and the other parameters. Linear regression analysis was applied. All reported p-values were based on two-sided tests and compared to a significance level of 0.05. The SPSS 18.0 for Windows (Chicago, IL, USA) was used for the analysis.

## 5. Conclusions

In conclusion, to the best of our knowledge, the present study is the first to investigate the correlation between glucose changes and changes in antioxidant markers and markers of apoptosis during an OGTT in children and adolescents with excess weight and the differences between children and adolescents with IGT and those with normal glucose tolerance. The response curves of the studied parameters in overweight children and children with obesity and IGT during the OGTT showed that glucose excursion was accompanied by a simultaneous increase in the antioxidant factors SOD and GPx3, which may have acted as a compensatory mechanism to prevent the impending increase in oxidative stress and apoptotic markers. Indeed, the studied apoptotic markers increased following the glucose rise and then decreased again, hypothetically under the influence of induced oxidative stress. Interestingly, in children with excess body weight and IGT, the antioxidant markers were more profoundly influenced (increased) by the glucose excursion during the OGTT compared to children with excess body weight and normal glucose tolerance, which may suggest a possible activation of protective compensatory mechanisms against further glucose increase in the IGT group. Further research is needed in order to confirm the hypothesis that the changes found in the oxidative status during the OGTT may reflect a more generalized activation of the antioxidant mechanisms in children with overweight or obesity as a preventative mechanism against glucose-mediated oxidative stress damage and obesity-related diseases. The possible involved pathogenetic mechanisms also need to be elucidated.

## Figures and Tables

**Figure 1 ijms-24-16517-f001:**
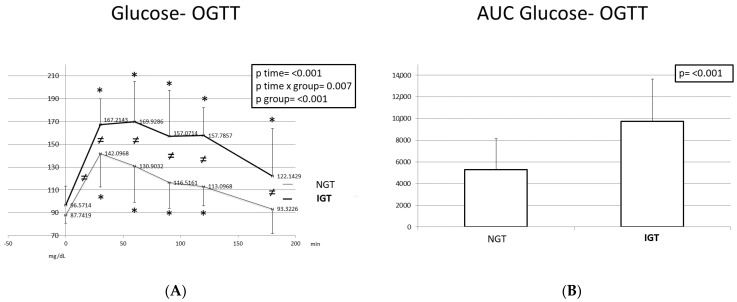
Glucose response (**A**) and glucose AUC (**B**) during the OGTT in participants with normal and disturbed OGTT. Mean glucose values are presented. Data were analyzed with ANOVA for repeated measurements, followed by a Bonferroni test for specific time points. *: Statistically significant difference (*p* < 0.05) between the different time points and the baseline. ≠: Statistically significant difference (*p* < 0.05) between the two studied groups at the same time point.

**Figure 2 ijms-24-16517-f002:**
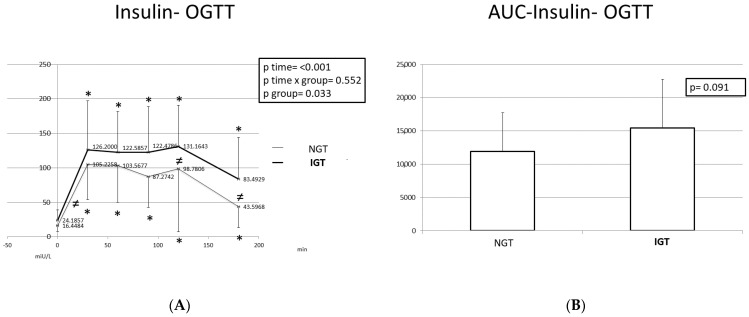
Insulin response (**A**) and insulin AUC (**B**) during the OGTT in participants with normal and disturbed OGTT. Mean insulin values are presented. Data were analyzed with ANOVA for repeated measurements, followed by a Bonferroni test for specific time points. *: Statistically significant difference (*p* < 0.05) between the different time points and the baseline. ≠: Statistically significant difference (*p* < 0.05) between the two studied groups at the same time point.

**Figure 3 ijms-24-16517-f003:**
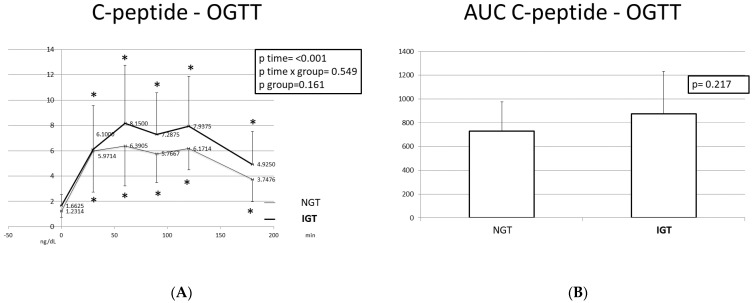
C-peptide response (**A**) and C-peptide AUC (**B**) during the OGTT in participants with normal and disturbed OGTT. Mean C-peptide values are presented. Data were analyzed with ANOVA for repeated measurements, followed by a Bonferroni test for specific time points. *: Statistically significant difference (*p* < 0.05) between the different time points and the baseline.

**Figure 4 ijms-24-16517-f004:**
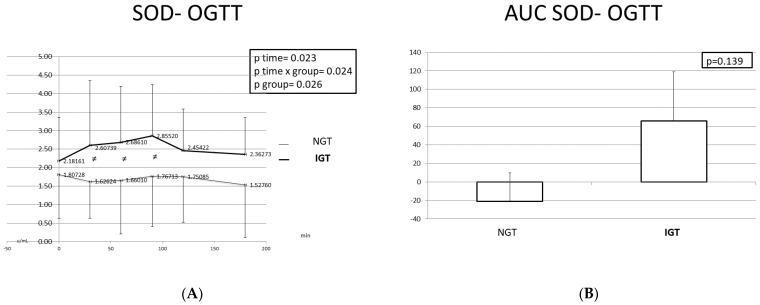
SOD response (**A**) and SOD AUC (**B**) during the OGTT in children with normal and disturbed OGTT. Mean SOD values are presented. Data were analyzed with ANOVA for repeated measurements, followed by a Bonferroni test for specific time points. ≠: Statistically significant difference (*p* < 0.05) between the two studied groups at the same time point.

**Figure 5 ijms-24-16517-f005:**
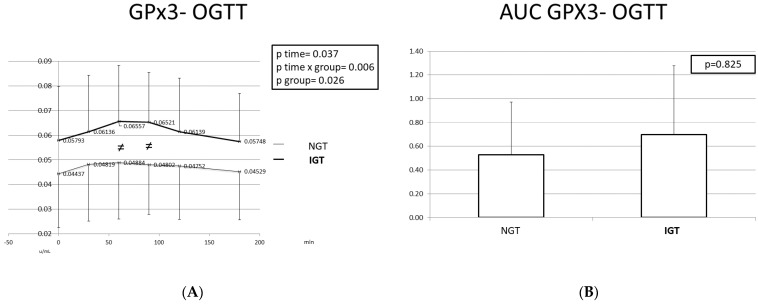
GPx3 response (**A**) and GPx3 AUC (**B**) during the OGTT in children with normal and disturbed OGTT. Mean GPx3 values are presented. Data were analyzed with ANOVA for repeated measurements, followed by a Bonferroni test for specific time points. ≠: Statistically significant difference (*p* < 0.05) between the two studied groups at the same time point.

**Figure 6 ijms-24-16517-f006:**
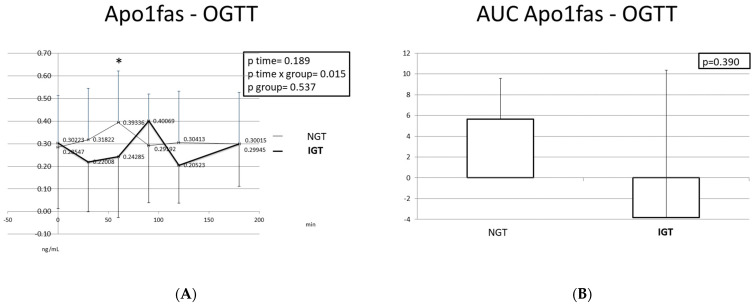
Apo1fas response (**A**) and Apo1fas AUC (**Β**) during the OGTT in children with normal and disturbed OGTT. Mean Apo1fas values are presented. Data were analyzed with ANOVA for repeated measurements, followed by a Bonferroni test for specific time points. *: Statistically significant difference (*p* < 0.05) between the different time points and the baseline.

**Figure 7 ijms-24-16517-f007:**
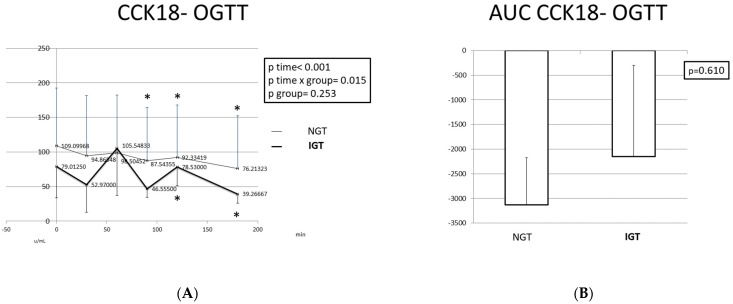
Cck18 response (**A**) and CCK18 AUC (**B**) during the OGTT in children with normal and disturbed OGTT. Mean cck18 values are presented. Data were analyzed with ANOVA for repeated measurements, followed by a Bonferroni test for specific time points. *: Statistically significant difference (*p* < 0.05) between the different time points and the baseline.

**Table 1 ijms-24-16517-t001:** Descriptive characteristics of children according to glucose disturbance (t test analysis).

Variable	Children without Glucose Disturbance N = 31	Children with Glucose Disturbance N = 14	*p*-Value
Age (years)	11.87 ± 2.5	12.79 ± 1.88	0.230
BMI%	92.19 ± 2.7	90.64 ± 3.9	0.750
Glucose (mg/dL)	87.74 ± 6.74	96.57 ± 16.72	0.015
Insulin (mIU/L)	16.45 ± 8.69	24.19 ± 14.87	0.033
HOMA-IR	65.41 ± 35.81	101.81 ± 56.91	0.012
Insulinogenic Index	1.94 ± 1.02	1.38 ± 0.72	0.071
C-peptide (ng/dL)	1.23 ± 0.5	1.67 ± 0.88	0.106
Triglycerides (mg/dL)	74.9 ± 37.13	84.29 ± 33.34	0.423
Cholesterol (mg/dL)	79.39 ± 35.38	176.43± 27.83	0.784
HDL (mg/dL)	53.53 ±15.83	48.64 ± 12.65	0.317
LDL (mg/dL)	114.19 ± 26.82	108.89 ± 21.69	0.534
T3 (ng/mL)	1.64 ± 0.42	1.86 ± 0.35	0.145
T4 (mg/dL)	9.09 ± 2.5	9.4 ± 1.18	0.686
TSH (IU/mL)	2.5 ± 3.3	2.25 ± 1.62	0.832
FT4 (ng/mL)	1.77 ± 0.85	1.6 ± 0.37	0.453
IGF-1 (ng/mL))	482.55 ± 314.72	642.08 ± 304.23	0.170
HbA1c (%)	5.12 ± 0.61	5.48 ± 0.73	0.131
Apolipoprotein a1 (mg/dL)	152.77 ± 41.67	149.7 ± 22.82	0.81
Apolipoprotein b (mg/dL)	80.36 ± 16.36	78.24 ± 16.12	0.73
Apolipoprotein b/Apolipoprotein a1	0.55 ± 0.14	0.53 ± 0. 12	0.688
Apo1fas (ng/mL)	0.29 ± 0.04	0.62 ± 0.032	0.144
cck18 (u/mL)	109.1 ± 14.97	75.26 ± 12.52	0.177
SOD (u/mL)	1.81 ± 0.22	2.18 ± 0.32	0.348
GPX3 (u/mL)	0.04 ± 0.04	0.058 ± 0.006	0.075
Leptin (ng/mL)	26.6 ± 3.89	18.29 ± 3.34	0.186
Ghrelin (mol/mL)	2.21 ± 0.19	2.72 ± 0.64	0.331
Adiponectin (ug/mL)	14.97 ± 1.04	10.23 ± 1.16	0.009

**Table 2 ijms-24-16517-t002:** Statistically significant correlations of the linear regression analysis in the studied markers in children with normal OGTT at the baseline.

		*p*-Value
GPx3	SOD	0.042
	Insulin	0.048
	HOMA-IR	0.040
	FT4	0.004
cck18 AUC	Insulin	0.021
	HOMA-IR	0.023
	HbA1c	0.002
Apo1fas	BMI%	−0.026
	Apolipoptorein-alpha	0.025
Apo1fas AUC	HbA1c	−0.025
	GPx3 AUC	0.043
SOD AUC	IGF-1	0.045

**Table 3 ijms-24-16517-t003:** Statistically significant correlations of the linear regression analysis in the studied markers in children with abnormal OGTT at the baseline.

		*p*-Value
GPx3	Insulin	0.018
	HOMA-IR	0.024
Apofas1 AUC	Apolipoprotein-alpha	0.002
	Insulin	−0.050
Fasting blood glucose	GPx3 AUC	0.001
	cck18 AUC	0.018
SOD AUC	HbA1c	0.038

## Data Availability

Data are unavailable due to privacy.

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
