# Peer review of "The Response of Antioxidant Enzymes and Antiapoptotic Markers to an Oral Glucose Tolerance Test (OGTT) in Children and Adolescents with Excess Body Weight"

_ijms, 2023, doi:10.3390/ijms242216517_

Round 1

Reviewer 1 Report

Comments and Suggestions for Authors

The authors evaluated the response of antioxidant enzymes and antiapoptotic markers to an Oral Glucose Tolerance Test (OGTT) in children and adolescents with excess body weight and concluded that the AUCs of glucose, insulin and c-peptide were greater in children with IGT. Glucose increase during an OGTT accompanied by a simultaneous increase of antioxidant factors, may reflect a compensatory mechanism against the impending increase of oxidative stress in children with IGT. Even though the sample size is small, the design was reasonable, and the results and conclusion were fair, and it would be a meaningful clinical guidance for the overweight management in children and adolescents after a big revision of the following problems.

1.     Please include the criteria for recruiting subjects in the Material and Methods. Any exclusion of serious disease or taking medications? It might affect the results.

2.     Why did the authors not include Hb1Ac parameter? It is more standard for evaluating the impaired glucose tolerance level.

3.     In Result section, please only note P value higher or lower than 0.05 after the statements. The exact number is not needed in the text since it has been listed in the table. In addition, when P value is lower than 0.05, it should consider no difference, not “a trend”.

4.     Why were P value and statistical difference not mentioned in the texts of 2.2 and 2.3?

5.     AUC of insulin and C-peptide in IGT is not statistically greater than NGT according to Figure 2 and Figure 3. So the conclusion should be revised.

6.      Please indicate what the * and # represent and compare with what in figure legends.

7.     According to the results, the error bars are really high which would fairly cover the significant difference. Did the authors do the sensitivity analysis to see whether any subject affected the results? The authors might consider doing a stratified analysis which might bring up more reasonable results and conclusions.

Comments on the Quality of English Language

Please read through the manuscript a few more times and if it is possible, it would be better if it can be revised by an English native speaker.

Author Response

Answers to the Reviewer  1

We would like to thank the reviewer for  the suggestions.

Comments and Suggestions for Authors

The authors evaluated the response of antioxidant enzymes and antiapoptotic markers to an Oral Glucose Tolerance Test (OGTT) in children and adolescents with excess body weight and concluded that the AUCs of glucose, insulin and c-peptide were greater in children with IGT. Glucose increase during an OGTT accompanied by a simultaneous increase of antioxidant factors, may reflect a compensatory mechanism against the impending increase of oxidative stress in children with IGT. Even though the sample size is small, the design was reasonable, and the results and conclusion were fair, and it would be a meaningful clinical guidance for the overweight management in children and adolescents after a big revision of the following problems.

  1.     Please include the criteria for recruiting subjects in the Material and Methods. Any exclusion of serious disease or taking medications? It might affect the results.

Thank you for your comment, this information has been added (line 428-430)

  1.     Why did the authors not include Hb1Ac parameter? It is more standard for evaluating the impaired glucose tolerance level.

HgbA1c was calculated in our population. The mean values are shown in table 1  while its correlations through linear  regression analysis is also noted. However, it was our oversight not to include it in  the material and methods. This information has been added (line 471).  Non- statistical significant differences were found for this parameter between the study  groups.

  1.     In Result section, please only note P value higher or lower than 0.05 after the statements. The exact number is not needed in the text since it has been listed in the table. In addition, when P value is lower than 0.05, it should consider no difference, not “a trend”.

Thank you for your comment. These  changes have been made in the text.

  1.     Why were P value and statistical difference not mentioned in the texts of 2.2 and 2.3?

Thank you for your comment, this information has been depicted in the figures which follow the text. A legend has been added after each figure in order to better understand the description of the symbols meaning.

  1.     AUC of insulin and C-peptide in IGT is not statistically greater than NGT according to Figure 2 and Figure 3. So the conclusion should be revised.

In the results section we have noted the following:  " The AUCs of glucose, insulin and c-peptide were greater in children with IGT, however only glucose differences were statistically significant".

  1. Please indicate what the * and # represent and compare with what in figure legends.

Thank you for the comment.  Your suggestion has been followed. This information has been added to each figure

  1.     According to the results, the error bars are really high which would fairly cover the significant difference. Did the authors do the sensitivity analysis to see whether any subject affected the results? The authors might consider doing a stratified analysis which might bring up more reasonable results and conclusions.

Thank you for your comment and we agree. However, the number of children in the IGT group is not large enough to allow stratified analysis. We would also like to note  that correlation analysis was performed after taking into consideration the confounding parameters of  gender and  age. This information has been added in the material and methods section. Children with outliers in measurements have been excluded from the study.  

Comments on the Quality of English Language

Please read through the manuscript a few more times and if it is possible, it would be better if it can be revised by an English native speaker.

Thank you for the comment.   The English revision has now been performed by a native speaker.

Reviewer 2 Report

Comments and Suggestions for Authors

The authors present an interesting paper showing how different antioxidant and apoptosis markers/parameters change along an oral glucose tolerance test in children with normal or impaired glucose tolerance. The work as merit, the research was thorough, the theme is contemporary, and the work is well framed in the scope of the journal. The work has limitations, the main one being the impossibility of having a control group of children with normal weight, but that is not feasible from an ethical standpoint. Nonetheless, this work will appeal to researchers and health professionals working in the fields of obesity, metabolic syndrome and diabetes. English in the manuscript is not always perfect but is generally adequate. There are some issues with the paper, but I believe they are manageable and solvable; therefore, I find the paper interesting enough and suitable for publication after the revisions proposed.

Main comments:

My only main concern with the paper is the use of references in general, which I believe could be more up to date. Not that the studies referenced are not adequate or meaningful in the context used, but in a constantly updated and growing field such as the study of obesity, metabolic syndrome and diabetes having only a small minority of referenced studies being produced in the last decade is a bit puzzling, at least for someone who has not worked in the field for a while and is looking in now. Maybe an effort could be made to include more recent references.

Minor comments:

Line 53: “HOMA-IR” Homeostatic Model Assessment for Insulin Resistance? This should not be abbreviated the first time used in the text.

Line 64: “LDL” Define the first time in the text.

Line 68: “SOD…” Do not use abbreviations at the beginning of sentences. Screen the manuscript for other instances in which this occurs.

Line 84: “HDL” Again, define the first time in the text.

Figure 1 (and the rest of the figures): Fonts in the images should be larger. It is especially hard to see the significancy symbols, which are, of course especially important.

Line 198: “Figure 1-7: Data were analyzed with ANOVA for repeated measurements, followed 198 by a Bonferroni test for specific time points. 199

*: Statistically significant difference (p<0.05) between the different time points and 200 the baseline. 201

≠: Statistically significant difference (p<0.05) between the two studied groups at the 202 same time point.”

This should be in every image from 1-7, or in the data analysis section. Nor randomly placed here...

Line 212: “Linear regression analysis showed statistically significant negative correlations be-212 tween: (1) Apo1fas and BMI% (p=0.026); (2) Apo1fas AUC and HgbA1C (p=0.025). 213 Also, a positive correlation between: (3) Apo1fas and apolipoprotein-alpha (p=0.025); (4) 214 SOD and GPx3 (p=0.042); (5) GPx3 and insulin (p=0.048), HOMA-IR (p=0.04), and FT4 215 (p=0.004); (6) Apo1fas AUC and GPx3 AUC (p=0.043); (7) cck18 AUC and insulin 216 (p=0.021); HOMA-IR (p=0.023) and HgbA1C (p=0.002); and (6) SOD AUC and IGF-1 217 (p=0.045).”

This would be better presented as a table. Same for the other experimental group.

Line 374: “1.75 gr/kg” Please use only “g” for grams.

Line 412: “McCord and Fridovich” Please include an actual reference.

Line 424: “the method of Wendel (1980)” Please include an actual reference.

Line 442: “Baseline correlation analysis was performed between the studied markers and various other possible confounding parameters, such the gender and the age.  analysis was also made between the AUC of the studied markers and the other parameters. Linear regression analysis was applied.”

I believe this should be in the Statistical Analysis section, and not in this particular one.

Comments on the Quality of English Language

English in the manuscript is generally adequate.

Author Response

Answers to the Reviewer  2

We would like to thank the reviewer for the suggestions.

Comments and Suggestions for Authors

The authors present an interesting paper showing how different antioxidant and apoptosis markers/parameters change along an oral glucose tolerance test in children with normal or impaired glucose tolerance. The work as merit, the research was thorough, the theme is contemporary, and the work is well framed in the scope of the journal. The work has limitations, the main one being the impossibility of having a control group of children with normal weight, but that is not feasible from an ethical standpoint. Nonetheless, this work will appeal to researchers and health professionals working in the fields of obesity, metabolic syndrome and diabetes. English in the manuscript is not always perfect but is generally adequate. There are some issues with the paper, but I believe they are manageable and solvable; therefore, I find the paper interesting enough and suitable for publication after the revisions proposed.

Main comments:

My only main concern with the paper is the use of references in general, which I believe could be more up to date. Not that the studies referenced are not adequate or meaningful in the context used, but in a constantly updated and growing field such as the study of obesity, metabolic syndrome and diabetes having only a small minority of referenced studies being produced in the last decade is a bit puzzling, at least for someone who has not worked in the field for a while and is looking in now. Maybe an effort could be made to include more recent references.

We thank you for your comment.  The references have now been carefully revised and updated in depth and numerous references have been replaced by recent ones. 

 Minor comments:

Line 53: “HOMA-IR” Homeostatic Model Assessment for Insulin Resistance? This should not be abbreviated the first time used in the text.

The explanation of the abbreviation has now been added into the text.

Line 64: “LDL” Define the first time in the text.

LDL's definition has been added into the text.

 Line 68: “SOD…” Do not use abbreviations at the beginning of sentences. Screen the manuscript for other instances in which this occurs.

It has been corrected and also some other changes have been made, according to your comment.

 Line 84: “HDL” Again, define the first time in the text.

HDL's definition has been added into the text. HDL has been defined the first time it is mentioned

Figure 1 (and the rest of the figures): Fonts in the images should be larger. It is especially hard to see the significancy symbols, which are, of course especially important.

Thank you for your comment.  All the  figures have now been modified according to your comment.

Line 198: “Figure 1-7: Data were analyzed with ANOVA for repeated measurements, followed 198 by a Bonferroni test for specific time points. 199

*: Statistically significant difference (p<0.05) between the different time points and 200 the baseline. 201

≠: Statistically significant difference (p<0.05) between the two studied groups at the 202 same time point.”

This should be in every image from 1-7, or in the data analysis section. Nor randomly placed here...

Thank you for your comment. All the figures have been  now been modified to reflect the corrections suggested.

 Line 212: “Linear regression analysis showed statistically significant negative correlations be-212 tween: (1) Apo1fas and BMI% (p=0.026); (2) Apo1fas AUC and HgbA1C (p=0.025). 213 Also, a positive correlation between: (3) Apo1fas and apolipoprotein-alpha (p=0.025); (4) 214 SOD and GPx3 (p=0.042); (5) GPx3 and insulin (p=0.048), HOMA-IR (p=0.04), and FT4 215 (p=0.004); (6) Apo1fas AUC and GPx3 AUC (p=0.043); (7) cck18 AUC and insulin 216 (p=0.021); HOMA-IR (p=0.023) and HgbA1C (p=0.002); and (6) SOD AUC and IGF-1 217 (p=0.045).”

This would be better presented as a table. Same for the other experimental group.

Thank you for the comment. The results are now presented as tables.

 Line 374: “1.75 gr/kg” Please use only “g” for grams.

It has been corrected

 Line 412: “McCord and Fridovich” Please include an actual reference.

The actual reference has been added

Line 424: “the method of Wendel (1980)” Please include an actual reference.

Actual reference has been added

Line 442: “Baseline correlation analysis was performed between the studied markers and various other possible confounding parameters, such the gender and the age.  analysis was also made between the AUC of the studied markers and the other parameters. Linear regression analysis was applied.”

I believe this should be in the Statistical Analysis section, and not in this particular one.

It has been removed and placed in the Statistical Analysis section of the  Methods section.

Comments on the Quality of English Language

The English in the text has  now been revised by a native speaker.

Reviewer 3 Report

Comments and Suggestions for Authors

General comments

The topic is unique and worthy of researching, as this review aimed to investigate the antioxidant and apoptotic markers response to an oral glucose tolerance test (OGTT) in a population of overweight children and adolescents, with normal (NGT) or impaired glucose tolerance (IGT) and to study the differences between children and adolescents with impaired and those  with normal glucose tolerance. The deduced conclusions based on the research methods/cases are enough and tenable. Regarding, the progress that had been made compared with the current research results; this Review highlights that the changes found in the oxidative status during the OGTT may reflect a more generalized activation of the antioxidant mechanisms in children with overweight or obesity as a preventative mechanism against glucose-mediated oxidative stress damage and obesity-related diseases. The possible involved pathogenetic mechanisms also need to be elucidated.                                                    

Strengths and weaknesses

The abstract is informative and reflect the body of the paper. The introduction provides sufficient background information for readers in the immediate field to understand the problem/hypotheses. The text is not well arranged or logic, the related concepts introduced clearly and the readability is sufficient. The discussion and theoretical analysis in this article is good. The reference section is informative and accurate.                                                                  

Suggestions for improvement

Paper is good, but needs the following:

-        The text is not well arranged

Comment: must be rearranged (Introduction and aim, Material and methods, Results, discussion, Conclusion then references)

-        Line 357:Subjects and ethics: 

Forty-five (45) children and adolescents, aged 7 to 16 years, were recruited for this observational study from the Outpatient Clinic.

 Comment: Regarding sex of subjects, all of them were boys or girls or both. What about sex difference in this study?

-        Line 367:  Antropometric measurements,

 Comment: Methods used must be described in details regarding measurement and BMI classification to overweight, obese or normal.

Comments on the Quality of English Language

minor revision

Author Response

Answers to the Reviewer  3

We would like to thank the reviewer for the suggestions.

Comments and Suggestions for Authors

General comments

 The topic is unique and worthy of researching, as this review aimed to investigate the antioxidant and apoptotic markers response to an oral glucose tolerance test (OGTT) in a population of overweight children and adolescents, with normal (NGT) or impaired glucose tolerance (IGT) and to study the differences between children and adolescents with impaired and those  with normal glucose tolerance. The deduced conclusions based on the research methods/cases are enough and tenable. Regarding, the progress that had been made compared with the current research results; this Review highlights that the changes found in the oxidative status during the OGTT may reflect a more generalized activation of the antioxidant mechanisms in children with overweight or obesity as a preventative mechanism against glucose-mediated oxidative stress damage and obesity-related diseases. The possible involved pathogenetic mechanisms also need to be elucidated.                                                    

Strengths and weaknesses

The abstract is informative and reflects the body of the paper. The introduction provides sufficient background information for readers in the immediate field to understand the problem/hypotheses. The text is not well arranged or logic, the related concepts introduced clearly and the readability is sufficient. The discussion and theoretical analysis in this article is good. The reference section is informative and accurate.                                                                  

Suggestions for improvement

Paper is good, but needs the following:

-        The text is not well arranged

Comment: must be rearranged (Introduction and aim, Material and methods, Results, discussion, Conclusion then references)

Thank you for your comment.  The manuscript, though, is written in the format required by  the  journal according to which it is mandatory to place the material and methods at the end of the manuscript.

-        Line 357: Subjects and ethics:

Forty-five (45) children and adolescents, aged 7 to 16 years, were recruited for this observational study from the Outpatient Clinic.

Comment: Regarding sex of subjects, all of them were boys or girls or both. What about sex difference in this study?

-Thank you for your comment.  This information has now been added into the text (27 boys and 18 girls). Baseline correlation analysis was performed between the studied markers and other possible confounding parameters, specifically the gender and age.

-        Line 367:  Anthropometric measurements,

 Comment: Methods used must be described in details regarding measurement and BMI classification to overweight, obese or normal.

Thank you for your comment. This information has now been added in the material and methods  section.

Comments on the Quality of English Language

minor revision

The English in the text has  now been revised by a native speaker.

Round 2

Reviewer 1 Report

Comments and Suggestions for Authors

1. It's Hb1Ac, not Hgb1Ac. Please revise the text and table 1.

2. Please also revise the section in Abstract accordingly, by adding "however only glucose differences were statistically significant" behind The area under the curve (AUC) of glucose, insulin and c24 peptide was greater in children with IGT.

Author Response

  1. It's Hb1Ac, not Hgb1Ac. Please revise the text and table 1.

Thank you for the comment.  Your correction  has been done

  1. Please also revise the section in Abstract accordingly, by adding "however only glucose differences were statistically significant" behind The area under the curve (AUC) of glucose, insulin and c24 peptide was greater in children with IGT.

Thank you for your comment. These  changes have been made in the abstract